# Trends in Heart Transplantation and Outcome Analysis: Nationwide Study Using the National Inpatient Sample and Readmission Database

**DOI:** 10.3390/medsci13020046

**Published:** 2025-04-22

**Authors:** Vivek Joseph Varughese, Aditya Sunil Bhaskaran, Hadrian Hoang-Vu Tran, Nikita Wadhwani, Vignesh Krishnan Nagesh, Izage Kianifar Aguilar, Damien Islek, Simcha Weissman, Adam Atoot

**Affiliations:** 1Department of Internal Medicine, University of South Carolina, Prisma Health, 2 Med Park, Richland, WA 29203, USA; 2Department of Internal Medicine, Lincoln Medical Center, New York, NY 10451, USA; sunilbhaskaranaditya@gmail.com; 3Department of Internal Medicine, Hackensack Palisades Medical Center, North Bergen, NJ 07047, USA; nikita.wadhwani@hmhn.org (N.W.); vgneshkrishnan@gmail.com (V.K.N.); izage.kianifaraguilar@hmhn.org (I.K.A.); damien.islek@hmhn.org (D.I.); simcha.weissman@hmhn.org (S.W.); adam.atoot@hmhn.org (A.A.)

**Keywords:** heart transplantation, heart failure, donor availability, mechanical circulatory support, left ventricular assist device, ECMO

## Abstract

Background: Heart transplantation (Htx) remains the definitive therapy for patients with end-stage heart failure. Despite advancements in mechanical circulatory support (MCS), immunosuppressive strategies, and organ allocation policies, donor availability remains a major limitation. This study analyzes the trends in Htx in the United States between 2016 and 2022, focusing on demographic shifts, mortality trends, and 30-day readmission patterns. Methods: We utilized the National Inpatient Sample (NIS) from 2016 to 2022 and the National Readmissions Database (NRD) for 2021 to identify Htx admissions using ICD-10 PCS code O2YA0Z0. Patient characteristics, mortality rates, and readmission patterns were analyzed using ANOVA and multivariate logistic regression, with statistical significance defined as *p* < 0.05. Results: The total number of Htx procedures increased from 641 in 2016 to 773 in 2022. The mean age of transplant recipients remained between 45 and 50 years, with no significant differences across years. Racial and socioeconomic disparities persisted, with approximately 60% of transplants occurring in White patients and 21–26% of recipients belonging to the lowest income quartile. All-cause in-hospital mortality remained stable at 4–7%. The 30-day readmission rate in 2021 was 57.7%, with heart failure, transplant rejection, and infections being the leading causes. Peripheral vascular disease (PVD) was the only comorbidity significantly associated with higher 30-day readmission risk (OR: 1.815, 95% CI: 1.477–2.230). Conclusions: Htx utilization has increased over time, driven by improvements in donor allocation and perioperative management. However, racial and socioeconomic disparities remain, and readmission rates continue to be high. Future efforts should focus on optimizing post-transplant care and addressing disparities to improve long-term outcomes.

## 1. Introduction

Heart transplantation (Htx) remains the last therapeutic resource for patients with end-stage advanced heart failure (HF). In 2018, UNOS changed the organ allocation policy for HTx. This was aimed at prioritizing patients with severe clinical conditions and, thereby, resulted in a reduction in mortality of people on the waiting list. Advanced heart failure and resistant angina are the current main indications for Htx, while active infections, peripheral vascular diseases, active malignancy, and morbid obesity are the contraindications [1]. Recent advancements in the realm of immunosuppressive medications have also contributed to better outcomes in patients undergoing Htx. Although 2023 marked the year with the highest number of transplants performed in the United States (US), donor availability still remains the main challenge in the expansion of this therapeutic strategy. Recently, there have been significant advances in organ allocation, donor–recipient matching, organ preservation, and expansion of the donor pool [2]. Innovations in the field of mechanical circulatory support (MCS) systems, including left ventricular assist device (LVAD) technologies, have further increased the scope of Htx. Although initially used as a bridge to Htx, the use of LVAD technologies as a destination therapy has been widely utilized, especially since the advent of the HeartMate II systems [3]. Advancement in the field of extracorporeal membrane oxygenation (ECMO) has also contributed to better patient outcomes by the advanced cardiopulmonary bypass these systems can provide in the recovery phase [4]. Beyond allocating donors, other factors that play a critical role in Htx include ABO/HLA compatibility, appropriate matching of graft size, ischemic time involved, age, and comorbid infections present at the time of transplantation [1]. The increased use of hearts from donors after circulatory death (DCD) has expanded the donor pool and has contributed to a rise in transplant numbers over the years [5]. The use of hearts from donors with hepatitis C virus (HCV) infection has also become more common, facilitated by effective antiviral treatments that mitigate the risk of HCV transmission to recipients [6].

In this study, we utilized the National Inpatient Sample (NIS) and National Readmissions Database (NRD) to analyze the trends in Htx in the US between 2016 and 2022. NIS and NRD are the largest publicly available identified patient databases available in the US.

## 2. Methods

The NIS for the years 2016 to 2022 and NRD for the year 2021 were used for the analysis. All statistical analyses were performed using STATA 18.5 MP. ICD 10 PCS code O2YA0Z0 was used to select admissions for Htx in the NIS and the NRD. Population stratification was performed. One-way ANOVA was used to stratify the differences in patient characteristics over the years (Kolmogorov–Smirnov test). A two-tailed *p* value < 0.05 was used to determine statistical significance of differences across the years. Index admissions requiring Htx were selected in the NRD, and 30-day readmission events were analyzed. Index admissions for Htx were stratified based on the readmissions in the 60 days following discharge. Patient factors including socioeconomic factors as well comorbidities were studied in both groups. Variance was analyzed using ANOVA. Factors that held significant association with the requirement of readmission in the 60 days following discharge in the ANOVA were used in multivariate logistic regression (probit model). Patient factors that held significant association with readmissions were analyzed, with statistical significance determined by a two-tailed *p* value < 0.05.

## 3. Results

Overall trends in Htx between the years 2016 and 2022 were analyzed using the NIS for the respective years. Population data are summarized in Table 1.

A general upward trend was observed in the total number of heart transplantations (Figure 1). A total of 773 heart transplantations were documented in 2022 compared to 641 in 2016.

The mean age of admissions for Htx shows a consistent trend over the years (Figure 2). The mean age of Htx admissions has remained between 45 and 50 years, with 95% confidence. ANOVA across the years showed no difference in mean age for Htx admissions over the years.

Analyzing Figure 3, the admissions for Htx that belonged to the lowest quartile of monthly income remained consistent over the years. The percentage has remained between 21% and 26%, with 95% confidence. Racial stratification did not show any significant variance across the years.

Figure 4 depicts the trends in all-cause mortality associated with Htx admissions between 2016 and 2022. No difference in all-cause mortality was observed over the years in the ANOVA test (*p* value: 0.752).

Using the National Readmission Database 2021, we identified patients who underwent a heart transplant between the months of January and November of 2021 and analyzed the readmission events in the 30 days following discharge of the index admission event. We identified 2775 index admission events for heart transplantation. Of this patient population, 1601 readmission events were recorded in the 30 days following discharge of the initial admission. Major readmission causes are summarized in Figure 5.

Multivariate regression analysis was performed to analyze factors that held significant association with 30-day readmissions following Htx. Regression analysis was performed accounting for age, sex, race, mortality predictive indices during the admission, and comorbid conditions that held significant variance in the ANOVA. A two-tailed *p* value < 0.05 was used to determine statistical significance. A documented diagnosis of peripheral vascular disease (OR: 1.815 95% CI: 1.477–2.230) was found to have a significant association with 30-day readmission following Htx. The results are summarized in Figure 6.

## 4. Discussion

With advancements in MCS and organ allocation strategies, the use of heart transplantation as a therapeutic strategy in the management of advanced heart failure has been showing an upward trend over the years (Table 1, Figure 1). Since 2021, the number of patients awaiting heart transplant at the beginning and end of the year has declined, despite more additions to the list than ever; this is primarily due to transplants because removals for death or “Other” reasons have declined, while removals for improvement, being too ill, or patient declining have remained stable. In 2020, there was a decrease in new additions to the waiting list, most likely due to the pandemic; however, since then, the number of new additions has increased by 1064 (26.6%). Between 2013 and 2023, the number of candidates awaiting transplant on December 31 of a given year decreased 15.2% [6]. With the evolving technologies at donor allocation and matching coupled with advancements in the realm of immunosuppressive medications improving outcomes, the trend is expected to increase over the coming years.

In our analysis, the mean age of patients undergoing Htx has remained consistently between 45 and 50 over the years (Figure 2). The American Society of Transplant Physicians and the National Institutes of Health have stated that age alone should not be considered a contraindication for heart transplantation, although patients over the age of 50 years may warrant additional screening for comorbid conditions [7]. The American Heart Association has traditionally considered age 55 years as an upper limit but acknowledges that carefully selected patients older than 55 years can successfully undergo heart transplantation [8]. Most centers do not select candidates above 65 years of age for Htx [9]. In a retrospective analysis by Awad et al. [10], similar 1-, 5-, and 10-year outcomes were observed in Htx recipients over 70 years of age when compared to younger recipients. Similar results were observed in two other retrospective studies [11,12]. While there is no absolute age cutoff, patients up to 70 years old are generally considered for heart transplantation, and those older than 70 years may be considered on a case-by-case basis, with careful selection criteria. There is a general paucity in the guidelines for definition of comorbidities that drive the decision for Htx in the elderly. With more data coming in suggestive of favorable outcomes in elderly patients, the mean age for heart transplantation candidates is expected to rise over the years. Of the total number of heart transplantations between 2016 and 2022, 23.12% of the transplants were for admissions with a documented diagnosis of congenital heart disease. The prevalence of pulmonary hypertension in the population that underwent transplantation was 9.23%. Despite the growing advancements in technologies as well as mechanical circulatory support systems, the all-cause in-hospital mortality has remained stable over the years. Major reasons postulated for this trend are the increasing number of transplant rejections in the population.

Racial disparities in heart transplantation are well documented and multifaceted, affecting both access to transplantation and post-transplant outcomes. Analyzing Figure 3, it is observed that around 60% of patients undergoing heart transplantation are white, and this trend has remained consistent over the years. Black and Hispanic patients are less likely to undergo heart transplantation compared to White patients. Despite an increase in the proportion of Black and Hispanic patients listed for transplantation, Black patients still have a lower likelihood of being transplanted, even with the new allocation system [13,14]. Additionally, Black patients are more likely to be listed with higher urgency statuses, yet they face higher waitlist mortality compared to White patients [15]. Hispanic recipients are more likely to experience failure to rescue (FTR), defined as the inability to prevent mortality after postoperative complications, compared to White recipients. However, Black recipients do not show significant differences in FTR rates but have lower overall survival [16]. The combination of donor and recipient race also impacts outcomes. Black recipients, regardless of donor race, have decreased survival compared to White recipients [17]. The percentage of women who receive Htx has remained below 35% consistently between 2016 and 2022 (Figure 3). Women are less likely to be referred for advanced heart failure therapies, including heart transplantation, compared to men. This may be due to implicit biases, differences in disease presentation, and concerns about sensitization [18,19]. Women experience higher waitlist mortality, particularly at the highest urgency status. This is partly due to adverse events associated with temporary mechanical circulatory support devices [20]. While post-transplant survival rates are generally similar between men and women, women are more likely to die from rejection and primary graft failure, whereas men are more likely to die from malignancies [21,22].

All-cause mortality for Htx had a consistent trend in our analysis (Figure 4), with the rates falling within 4–7% across the years, with 95% confidence. Analyzing the literature, mortality trends associated with heart transplantation have been improving over the past few decades. Recent data from the OPTN/SRTR 2022 Annual Data Report, published by the American Society of Transplantation and the American Society of Transplant Surgeons, indicate that post-transplant mortality has been stable to slightly better since 2011. For adult recipients who underwent transplants between 2015 and 2017, the 1-, 3-, and 5-year survival rates were 91.3%, 85.7%, and 80.4%, respectively [6]. A study published in The Annals of Thoracic Surgery in 2021 found that the 10-year survival rate for heart transplant recipients was 53%, with a standardized mortality rate (SMR) of 2.84 compared to the general population. This study also noted that long-term mortality rates have consistently declined over time [23].

The 30-day readmission rate following Htx was 57.69% (1601 readmission events in 2775 index admissions). Heart failure (HF) was noted to be the major readmission cause in the 30 days following transplant, with 522 readmission events documented. Transplant rejection and infections were other major readmission causes in the 30-day period. In a retrospective analysis performed by Campos et al. [24], the 90-day readmission rate following Htx was noted to be 49.4%, with infections and transplant rejection noted to be the major causes for readmission. In the multivariate regression analysis, peripheral vascular disease (PVD) was the only comorbidity found to have a significant association with 30-day readmission following Htx (OR: 1.815 95% CI: 1.477–2.230). A study analyzing data from the United Network of Organ Sharing (UNOS) database found that patients with pre-transplant symptomatic PVD had significantly lower 1-, 5-, and 10-year survival rates compared to those without PVD (91.5% vs. 94.9%, 74.8% vs. 82.6%, and 48.6% vs. 54.7%, respectively) [25]. Additionally, the presence of PVD can complicate the post-transplant period, as patients with PVD may require additional vascular interventions, such as revascularization or amputation, which can further impact their overall health and recovery [26]. The American Heart Association (AHA) also notes that the wide spectrum of peripheral and cerebrovascular disease makes it challenging to establish specific exclusion criteria for HT. However, the potential for acute thrombotic or embolic events, the need for postoperative intra-aortic balloon support, and the effects of corticosteroids on atherosclerotic progression are important considerations [8]. Although prior retrospective studies [27] have proved uncontrolled hypertension as an independent risk factor for readmissions following Htx, no significant association was seen in our analysis.

The total artificial heart (TAH) has been postulated as the future of advanced heart failure management and would be a potential solution for candidates not eligible for heart transplantation. Although being designed as a bridge to transplant, its use as a destination therapy, just like LVADs, is increasingly being considered [28]. The SynCardia TAH, the only FDA-approved TAH, is a pulsatile system capable of providing flows greater than 9 L/min. It is indicated for patients at imminent risk of death from non-reversible biventricular failure. The American Heart Association notes that the TAH is reserved for patients with severe biventricular failure who are not candidates for other mechanical circulatory support devices, like left ventricular assist devices (LVADs) [29].

## 5. Conclusions

There has been a general increasing trend in the total number of heart transplantations between 2016 and 2022. All-cause mortality associated with Htx has remained stable at 4–7% across the years. The percentage of females undergoing Htx has remained consistently below 35% between 2016 and 2022. Heart failure and related conditions, transplant rejection, and infections were the major admission causes in the 30 days following Htx. Peripheral vascular disease (PVD) held significant association with 30-day readmission following Htx.

## 6. Limitations of This Study

While the NIS and the NRD have utility in large population studies, these databases are not without limitations: one potential limitation is the reasons for the in-hospital mortality were not documented for heart transplant admissions, making it a limitation. The ICD and PCS codes used for patient selection and stratification are subject to inter-operator variability.

## Figures and Tables

**Figure 1 medsci-13-00046-f001:**
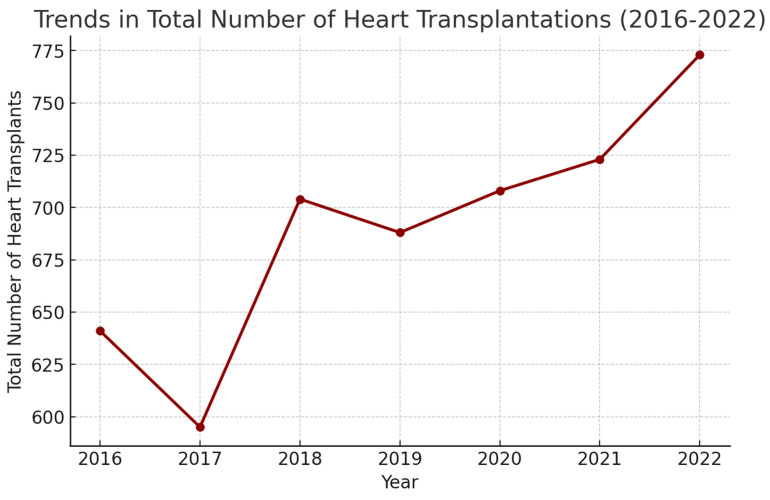
Trends in total number of heart transplantations (2016–2022).

**Figure 2 medsci-13-00046-f002:**
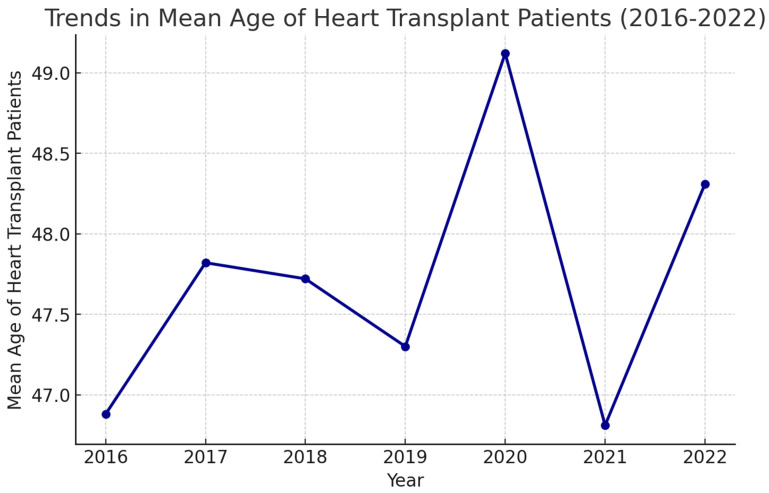
Trends in mean age of admissions undergoing heart transplantation.

**Figure 3 medsci-13-00046-f003:**
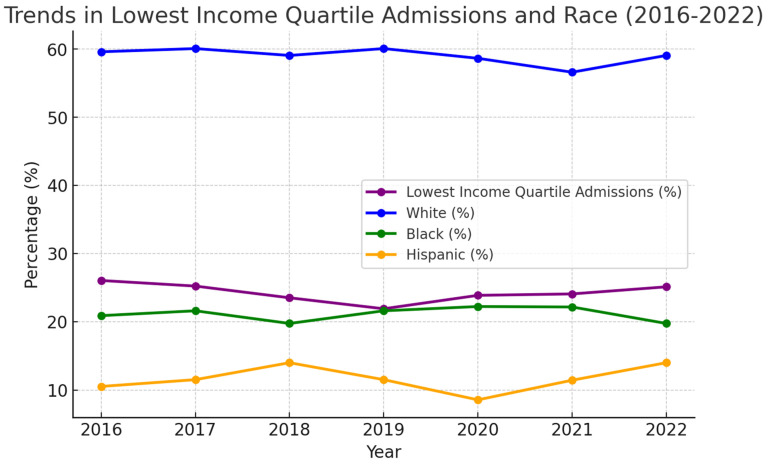
Stratification by race and income quartile for heart transplantation admissions (2016–2022).

**Figure 4 medsci-13-00046-f004:**
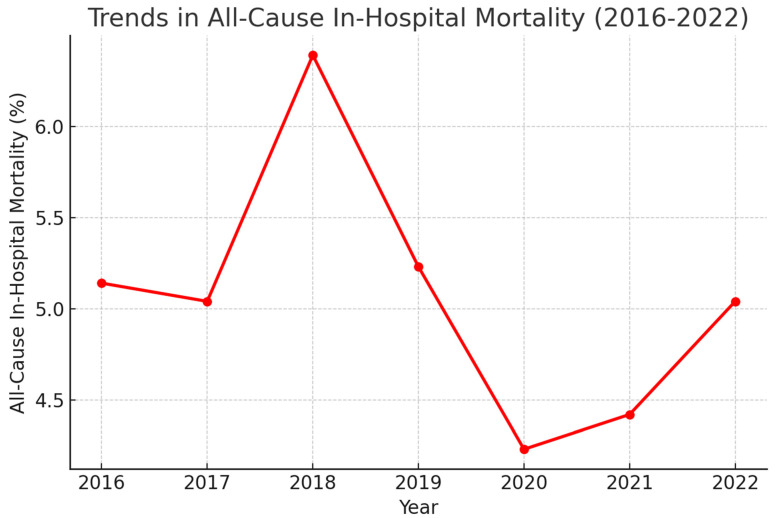
Trends in all-cause mortality for heart transplantation admissions (2016–2022).

**Figure 5 medsci-13-00046-f005:**
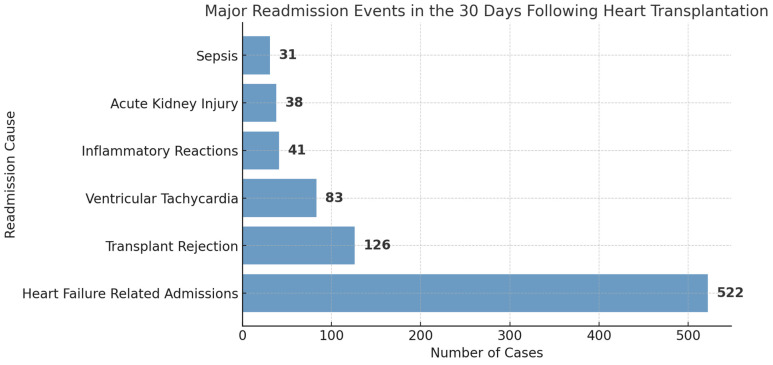
Major readmission causes in the 30 days following heart transplantation.

**Figure 6 medsci-13-00046-f006:**
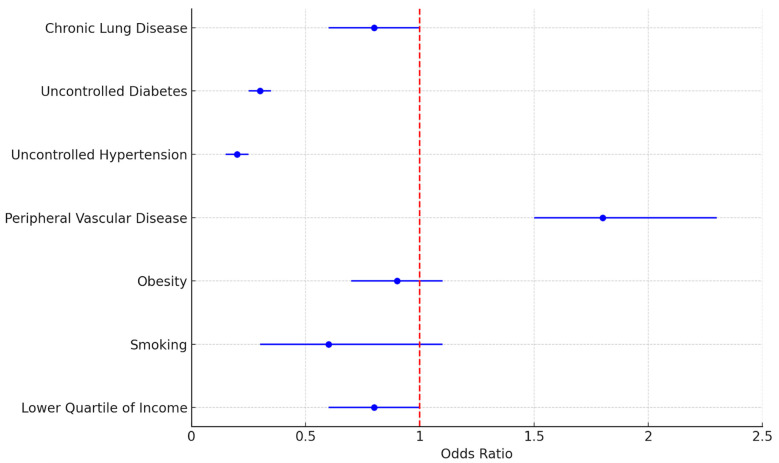
Association of patient factors with 30-day readmission following heart transplantation.

**Table 1 medsci-13-00046-t001:** Population trends in heart transplantations (2016–2022).

Year	Total Heart Transplants	Mean Age (95% CI)	Male (%)	Female (%)	Lowest Income Quartile Admissions (%)	White (%)	Black (%)	Hispanic (%)	All-Cause In-Hospital Mortality (%)
2016	641	46.88 (45.30–48.46)	73.32	26.68	26.03	59.6	20.87	10.52	5.14 (3.86–7.15)
2017	595	47.82 (46.25–49.40)	72.61	27.39	25.21	60.07	21.58	11.51	5.04 (3.54–7.12)
2018	704	47.72 (46.21–49.23)	67.67	32.24	23.49	59.06	19.73	13.99	6.39 (4.80–8.45)
2019	688	47.30 (45.85–48.75)	73.32	26.68	21.88	60.07	21.58	11.51	5.23 (3.79–7.15)
2020	708	49.12 (47.71–50.53)	73.87	26.13	23.85	58.64	22.21	8.56	4.23 (2.97–5.99)
2021	723	46.81 (45.31–48.31)	73.87	26.68	24.06	56.6	22.14	11.43	4.42 (3.14–6.19)
2022	773	48.31 (46.95–49.67)	73.32	26.87	25.1	59.06	19.73	13.99	5.04 (3.70–6.83)

## Data Availability

Analysis was performed using HCUP NRD/NIS: deidentified patient data from public database: no IRB approval or consents required.

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
