# Peer review of "Trends in Heart Transplantation and Outcome Analysis: Nationwide Study Using the National Inpatient Sample and Readmission Database"

_medsci, 2025, doi:10.3390/medsci13020046_

Round 1
Reviewer 1 Report
Comments and Suggestions for Authors
Commentaries are also in attach

Author Response
They discussed some aspects of
trends in heart transplant outcomes for patients over the age of 70 years and in the lowest
urgency candiadtes, but only generally and not with the data of the research. Therefore, I
suggest a deeper analysis of those subgroups and others with interest like diabetics, patients
under mechanical support, congenital heart disease, pulmonary heart disease with need of
concomitant lung transplantation.
REPLY: We could not identify patients > 70 years of age in the database. Prevalence of pulmonary hypertension and congenital heart diseases in the transplant population has been added.
. The survival rate has not changed over the years. So, what could explain this aspect although
a lot of advances have been seen in the field of immunosuppression and follow-up?
REPLY: have added in the discussion section that the increasing amount of transplant rejections could be a potential cause for this
Reference has been added for artificial heart.
Reviewer 2 Report
Comments and Suggestions for Authors
This work is interesting. However, authors need to discuss in detail in the discussion section why there is such a discrepancy in social and ethnic minorities undergoing transplants
Author Response
This work is interesting. However, authors need to discuss in detail in the discussion section why there is such a discrepancy in social and ethnic minorities undergoing transplants
REPLY: has been added in the discussion
Racial disparities in heart transplantation are well-documented and multifaceted, affecting both access to transplantation and post-transplant outcomes. Analyzing Figure 3, it was observed that around 60% of patients undergoing heart transplantation are white, and this trend has remained consistent over the years. Black and Hispanic patients are less likely to undergo heart transplantation compared to White patients. Despite an increase in the proportion of Black and Hispanic patients listed for transplantation, Black patients still have a lower likelihood of being transplanted even with the new allocation system [13, 14]. Additionally, Black patients are more likely to be listed with higher urgency statuses, yet they face higher waitlist mortality compared to White patients [15]. Hispanic recipients are more likely to experience Failure to Rescue (FTR), defined as the inability to prevent mortality after postoperative complications, compared to White recipients. However, Black recipients do not show significant differences in FTR rates but have lower overall survival [16]. The combination of donor and recipient race also impacts outcomes. Black recipients, regardless of donor race, have decreased survival compared to White recipients [17]. The percentage of women who receive Htx have remained below 35% consistently between 2016 and 2022 ( Figure 3). Women are less likely to be referred for advanced heart failure therapies, including heart transplantation, compared to men. This may be due to implicit biases, differences in disease presentation, and concerns about sensitization [18,19].Women experience higher waitlist mortality, particularly at the highest urgency status. This is partly due to adverse events associated with temporary mechanical circulatory support devices [20].
Reviewer 3 Report
Comments and Suggestions for Authors
The authors used two databases, National Inpatient Sample (NIS) and National Readmissions Database (NRD) for 2021, to conduct a brief descriptive study of heart transplant patients during this period.
1. The authors only listed the number of surgeries, the mean age, all-cause in-hospital mortality, and the racial and socioeconomic status, without further analysis of the underlying information that causes the mortality rate, and the information provided in the article was too thin.
2. the author mentioned in Multivariate regression analysis that Peripheral Vascular Disease was found to have a significant association with 30 day readmission following Htx, there is no further discussion of this risk factor.
3. In the section looking to the future, the potential impact of emerging technologies on heart transplantation is not fully considered. With the development of technology, new techniques may change the status quo of heart transplantation, and including the discussion can make the research more forward-looking.
Author Response
The authors only listed the number of surgeries, the mean age, all-cause in-hospital mortality, and the racial and socioeconomic status, without further analysis of the underlying information that causes the mortality rate, and the information provided in the article was too thin.
REPLY: a very valid point that will be mentioned in the limitations section: the database does not let us know the reason for the mortality: going by the diagnoses codes at discharge for patients who died, might be misleading as we cannot be sure which recorded diagnoses was the actual cause of death
the author mentioned in Multivariate regression analysis that Peripheral Vascular Disease was found to have a significant association with 30 day readmission following Htx, there is no further discussion of this risk factor.
REPLY: has been added in the discussion section
"Peripheral Vascular Disease (PVD) was the only comorbidity found to have a significant association with 30 day readmission following Htx ( OR: 1.815 95% CI: 1.477 - 2.230). A study analyzing data from the United Network of Organ Sharing (UNOS) database found that patients with pre-transplant symptomatic PVD had significantly lower 1-, 5-, and 10-year survival rates compared to those without PVD (91.5% vs. 94.9%, 74.8% vs. 82.6%, and 48.6% vs. 54.7%, respectively) [25]. Additionally, the presence of PVD can complicate the post-transplant period, as patients with PVD may require additional vascular interventions, such as revascularization or amputation, which can further impact their overall health and recovery [26]. The American Heart Association (AHA) also notes that the wide spectrum of peripheral and cerebrovascular disease makes it challenging to establish specific exclusion criteria for HT. However, the potential for acute thrombotic or embolic events, the need for postoperative intra-aortic balloon support, and the effects of corticosteroids on atherosclerotic progression are important considerations "
In the section looking to the future, the potential impact of emerging technologies on heart transplantation is not fully considered. With the development of technology, new techniques may change the status quo of heart transplantation, and including the discussion can make the research more forward-looking.
REPLY: extra section on artificial heart and future trends added
"
Total artificial heart (TAH) has been postulated as the future of advanced heart failure management and would be a potential solution for candidates not eligible for heart transplantation. Although being designed as a bridge to transplant, its use as a destination therapy, just like LVADS are increasingly being considered [29]. The SynCardia TAH, the only FDA-approved TAH, is a pulsatile system capable of providing flows greater than 9 L/min. It is indicated for patients at imminent risk of death from non-reversible biventricular failure. The American Heart Association notes that the TAH is reserved for patients with severe biventricular failure who are not candidates for other mechanical circulatory support devices like left ventricular assist devices (LVADs) [30]."
Round 2
Reviewer 3 Report
Comments and Suggestions for Authors
It is suggested that the author recheck the language coherence and formatting of the entire text.